# Investigation of Technological and Load Intensity Parameters of the Finishing Process of Materials on Equipment with Tools Translational Kinematics

**DOI:** 10.3390/ma15093048

**Published:** 2022-04-22

**Authors:** Karim Ravilevich Muratov, Timur Rizovich Ablyaz, Evgeny Anatolevich Gashev, Irina Georgievna Goryacheva

**Affiliations:** 1Mechanical Engineering Faculty, Perm National Research Polytechnic University, 614000 Perm, Russia; karimur_80@mail.ru (K.R.M.); kot_ostrow@mail.ru (E.A.G.); 2Institute for Problems in Mechanics of the Russian Academy of Sciences, IPMech RAS, 119526 Moscow, Russia; 11188@mail.ru

**Keywords:** abrasive lapping, hypotheses Preston, surface roughness, contact area, constant clamping force, variable clamping force

## Abstract

The regularities of the formation of the resulting raster tool trajectories based on Lissajous figures for the lapping process of planes are established. This makes it possible to maximize the cutting ability of the tool, which contributes to its more uniform wear and increased productivity and processing quality. Optimal parameters of productivity and roughness of the treated surface during lapping of zirconium ceramics are achieved through the use of ASM paste 28/20 µm. Based on Preston’s hypothesis, an exponential dependence of the change in the contact area during the lapping of planes of different initial shape of the macrorelief is obtained. The obtained theoretical and practical results of the study of the process of flat lapping with constant and variable clamping force of the treated surface to the surface of the tool. The influence of the force factor on the formation of the surface in the process of abrasive lapping has been established. Studies have been carried out and the main technological recommendations of precision surface treatment of workpieces based on hard, brittle ceramic material and bronze samples on equipment with a raster trajectory of the tool movement are presented. The optimal pressure value when processing ceramics should be considered 203–270 kPa (2.1–2.8 kg/cm^2^).

## 1. Introduction

One of the leading trends in industrial production has been and remains the growing need to improve quality, increase productivity, increase durability and reliability of devices and products, improve their presentation. It is known that technological indicators are provided to a greater extent at finishing operations due to surface quality management and the achievement of high performance characteristics of the machined parts in the final stage of the processing process. The long technological process of processing high-precision parts is completed by lapping operations, the purpose of which is to obtain the required roughness and high accuracy of the geometric shape of the surfaces being processed.

Ensuring high-quality surface processing of such hard, crunchy and wear-resistant materials as hardened steel, ceramics, glass, single crystals is a complex technological task. Along with the low roughness, limited by fractions of micrometers, a specific requirement is the absence of surface defects in the form of chips, microcracks, micro-punctures. In this regard, the role of the influence of technological factors increases, the selection of equipment, rational cutting modes, with the use of the appropriate material in the processing of hard, brittle and wear-resistant materials.

Abrasive lapping and polishing are the technological processes of the final finishing of high-precision parts after the operation of abrasive grinding [1]. Abrasive lapping is a labor-intensive finishing operation that allows to obtain surfaces with a roughness of *Ra* from 0.16 to 0.006 μm with geometric shape deviations up from 2 to 0.1–0.3 μm [2,3].

A characteristic feature of any lapping process should be recognized as low cutting speeds. The range of cutting speeds during finishing is 0.1–4 m/s. Therefore, the process of finishing with circles or bars can be attributed to micro-grinding [4], which is increasingly developing with the production of high-quality micro-powders and bundles.

When lapping plastic materials, i.e., most metals, the removal is carried out by abrasive grains fixed on the lapping and scratching the treated surface. When lapping brittle materials, on the contrary, free grains, rolling parts on the surface and puncturing material particles work effectively [4].

It is used in the final processing of measuring and cutting tools and machine parts with hardened, nitrided and carbide surfaces. It is also advisable to use lapping when processing brittle non-metallic materials and when it is necessary to preserve the special properties of the source material that change under thermal and force influences or to remove the defective layer of material formed during the pre-processing.

Depending on the type of tool, there are two types of abrasive finishing:Free abrasive lapping (pastes, suspensions)Processing with fixed abrasive (abrasive wheels, saturated lapp).

Lapping with a fixed abrasive is a promising process that provides more stable results in productivity and surface quality. Studies have shown that the intensity of saturated lapp of the surface of the part with an abrasive in this case sharply decreases [2].

The productivity and efficiency of lapping depends not only on the material and grain size of pastes and suspensions, but also on the main technological parameters of the process: the values of the lapp pressure on the part at the point of their contact, the speed of working movement and the time of lapping. The pressure value during the finishing process varies in the range of 30–600 kPa, depending on the type of material being processed. The finishing time varies in the range of 1–60 min.

According to the kinematic feature, most machines can be divided into two groups: machines with translational working motion and machines with rotational movement of the tool (lapp).

Machines with rotating lapp due to the simplicity of design and versatility have become more widespread. The main advantage of this kinematics is the possibility of providing increased processing speeds (2–4 m/s) with uniform distribution of abrasive material. The presence of lapping rotation negatively affects the accuracy of lapping due to different linear speeds of parts located at different distances from the center of lapp rotation. Therefore, the removal of metal from the parts and the wear of the lapp in different zones are not the same, which leads to significant size fluctuations in the batch of simultaneously processed parts. The disadvantages include uneven wear, as a result of the inequality of linear speeds and the use of only 1/3 of the tool diameter due to the zero speed in the center of the lapp. In addition, the disadvantages of these machines include the limited (inner diameter of the ruling rings) dimensions of the parts being adjusted and the high consumption of abrasive paste, suspension for fine-tuning the correct rings themselves.

It is established that the presence of a complex working movement, consisting of several movements of the tool (lapp) and the workpiece, contributes to improving the quality and productivity of lapping. When lapping, this type of kinematics has a beneficial effect on the efficiency of the process for the following reasons:(1)uniform wear of the working surface of the lapp is achieved, which increases the accuracy of the dimensions and shape of the machined parts;(2)abrasive grains are able to work with a large number of their vertices and edges, which leads to an increase in their cutting ability and durability;(3)the microrelief of the surface is leveled due to the grain cutting of protrusions of irregularities and filling of depressions with displaced metal;(4)a dense uniform grid of traces is applied to the treated surface, which helps to increase the actual bearing surface of the part, creates conditions for better retention of grease [5].

Machines with oscillatory (progressive) working movement of lapp include lappingmachines of the “Raster” model, developed at the Perm National Research Polytechnic University (PNRPU, Russia). A distinctive feature of the machines is the non-repeating trajectory of the working movement of lapp in the form of Lissajous figures of varying complexity [6].

Generalization of the published results of theoretical and experimental studies of the kinematics of lapping equipment allows us to formulate the following basic requirements for the resulting movement of tools and parts during the finishing abrasive treatment of high-precision surfaces:(1)the presence of complex movement of cutting grains;(2)equality of speeds of all points of the lapped surface and the tool;(3)stable reproducibility, regulation and the ability to control the resulting cutting trajectory.

In the existing models of lappingmachines, all these requirements are not fully met. Improving the methods of finishing abrasive treatment of precision surfaces due to controlled processes of lapping operations aimed at obtaining high accuracy of geometric shape, increasing the productivity of processing and the required microrelief of the treated surface is an acute problem.

The purpose of the study is to determine the rational types and parameters of the trajectory of the working tool movement and the development of equipment for its implementation. Determination of rational types and granularity of abrasive materials when lapping brittle and plastic materials. Analysis of the influence of constant and variable clamping forces and determination of optimal patterns of load changes during processing.

## 2. Material and Methods

The study of the process of abrasive lapping of samples from ceramics and bronze alloy was carried out at the Perm National Research Polytechnic University (PNRPU). The dimensions of the ceramic samples produced: width-5 mm, height-5.8 mm and length-44 mm. To study the influence of constant and variable forces on the dynamics of the formation of the flatness of the treated surface, bronze-based samples (CuSn5Zn5Pb5-C) were used. In these studies, a soft plastic material was used to reduce the time of research and clarify the general pattern of the influence of force factors on the formation of the surface during abrasive lapping for any lapped materials. A bronze specimen with a diameter of 40 mm, a height of 14 mm. According to the specifications, the accuracy of the geometric shape in the form of a deviation from flatness is no more than 1 μm, it was required to obtain.

Roughness measurements were carried out on a perthometer S2 by Mahr (Göttingen, Germany). The average value of the initial roughness of ceramic according to the parameter *Ra* is 4–5 microns.

The measurement of the initial deviation from straightness was carried out on a Mahr MMQ 400 round gauge. For various samples, the initial deviations from straightness ranged from 30 to 75 μm.

Experimental testing of the abrasive lapping of the surface of samples from ZrO_2_ ceramics was carried out on a lapping machine “Raster 220 (Perm, Russian)” with oscillatory (progressive) movement of the tool (lapp) and rotation of the treated surfaces relative to the center of the tool [7,8]. The machine is designed for finishing a variety of machine-building parts with precision flat surfaces. It allows you to process both individual parts and batches of parts in multi-seat fixtures, as well as to corrected lapp.

A prototype of a lappingmachine with a complex progressive kinematics of the tool is conventionally called “Raster 220”. This machine reproduces a complex adjustable movement of the tool (lapp) describing the Lissajous trajectories of various configurations.

The «Raster 220» machine is shown in Figure 1 and consists of three main components: a drive unit, a pressure device, a control panel with a frequency converter. The drive unit 1 contains an electric motor and transmissions that reduce the speed of rotation, and also includes a mechanism that converts rotation into an oscillatory motion of the lapping and a device for controlling its trajectory. The pressure device 3 is used to press the processed parts to the lapp 2 with an adjustable force and to communicate additional movement to the parts relative to the lapping due to the force of abrasive friction. The machine is equipped with a control panel 4, which houses an electrical circuit and a time relay that allows you to work in semi-automatic mode, and a frequency converter 5 that allows you to continuously adjust the frequency of lapping vibrations. Table 1 shows the basic numerical data of the machine.

The condition for uniform wear of the lapp is the uniform distribution of the processed parts over its surface. The fulfillment of this requirement is ensured by the use of appropriate devices for fixing the processed parts.

To fine-tune the working surface of samples from ZrO_2_ ceramics, a special device has been developed and manufactured that allows simultaneous processing of three parts. The universal device (Figure 2) consists of a body 1, in the grooves 2 of which the machined parts are installed 3. The machined parts are fixed by tightening the screw 4 with a clamp 5. To prevent axial movement of the parts, elastic limiters 6 are installed in the grooves, fixed with pins.

The machined parts are installed in the slots of the housing, are based on two perpendicular faces and are clamped. After that, the device, together with the processed parts, is installed on the lapping. Next, a washer 7 is placed on the device, which is centered along the inner diameter of the recess in the housing 1. Washer 7 is made of tool steel with hardness parameters HRC 63. For the manufacture of washer 7, the technology of electrical discharge machining was used. As electrode-tools was used new composition of the composite materials. New electrode-tools materials obtained within the framework of the grant of the Russian Science Foundation No. 20-79-00048. For the manufacture electrode-tools were used powders of electrolytic copper PMS-1, a preparation of dry colloidal graphite of grade S-1, thermally expanded graphite (TRG). Processing modes were assigned to ensure the specified quality indicators of the treated surface. The required working pressure of the machined parts on the lapping is created by the pressing device of the machine. This design of the device makes it possible to compensate for the different heights of the processed parts and to install them on the surface of the lapp. This ensures an even distribution of pressure on each sample.

Measurement and evaluation of deviations in the geometric shape of the treated surface of samples made of zirconium ceramics and the working surface of the lapping (tool) were carried out on a measuring machine MMQ 400 company Mahr (Göttingen, Germany). 

The deviation from the straightness of the working surface of the lapp was also measured using the MMQ 400 measuring machine. The evaluation of the deviation from the straightness of the lapping is carried out based on the results of two measurements in the orthogonal planes.

The roughness of the treated surface was evaluated in accordance with ISO 4287-1997 standard according to the following main criteria: *Rq*, *Ra*, *Rz*, *Rmax*, *Rt*. Roughness measurements were carried out on a profilograph profilometer S2 by Mahr (Göttingen, Germany) using a special probe with a support allowing to exclude the main external fluctuations.

To obtain reliable results in each series of experiments, the roughness parameters were measured on several samples. The average value of the parameters was automatically determined by five values measured within the base length. Another method of surface quality assessment is visual analysis using a MET 6/6S microscope manufactured by Altami (Saint-Petersburg, Russia). The magnification range, which ranges from ×50 to ×1000.

Research and analysis of optical, physical and mechanical properties of brittle non-metallic materials are often accompanied by great technological difficulties that require the development of non-standard, original methods. Thus, a team of researchers from the Perm State National Research University has developed a technique for monitoring and assessing the depth of the destroyed layer in a brittle optical single crystal based on lithium niobate [9,10,11].

To test the effectiveness of raster abrasive finishing with variable clamping force, a special technique was developed for determining the actual contact area of the treated surface of ceramic samples with lapp. This technique consists in determining the area (S_f_) of the surface of the parts in contact with the tool during the finishing process by scanning and subsequent image processing programmatically.

The methodology of visual analysis of the assessment of the actual contact area of the treated surface:(1)Before the experiment, a thin layer of special spray;(2)As a result of processing in the area of the contact spot of the part and the lapping, the paint was erased, leaving only the untouched areas tinted (Figure 3a);(3)The completed face of the sample is scanned;(4)Programmatically, the surface areas in contact with the lapping are separated from the untreated areas (Figure 3b) and the ratio of their areas as a percentage is calculated;(5)The results obtained as a percentage, taking into account the nominal area of the treated surface of the sample, are recalculated in cm^2^.

The evaluation of the processing performance of ceramic samples was carried out by measuring the volume removal of the material. The amount of volumetric removal of the material was determined as the difference in volumes before and after finishing. The volume of the parts was calculated by the formula: *V* = *m*/*p*, where m is the mass of the parts, *p* is the average density of the material of the parts. The mass of the parts was determined by weighing on the scales of laboratory electronic HR company “A&D Co.LTD” (Tokyo, Japan).

In the course of experiments on the surface finishing of samples from ZrO_2_ ceramics, the nomenclature of abrasive material was tested, differing in the type of abrasive and granularity:Synthetic diamond paste (ASM):

ASM 7/5 (grain size from 5 to 7 microns);

ASM 14/10 (grain size from 10 to 14 microns);

ASM 28/20 (grain size from 20 to 28 microns);

ASM 40/28 10 (grain size from 28 to 40 microns);

ASM 60/40 10 (grain size from 40 to 60 microns).

2.Paste cubic boron nitride (CNB):

CNB 28/20 (grain size from 20 to 28 microns); 

CNB 40/28 (grain size from 28 to 40 microns).

3.Silicon carbide powder green (KZ):

KZ M28 (grain size 28 microns);

KZ M40 (grain size 40 microns).

Paste and powder were used in the experiments. The weight of the paste was 0.150 g (0.05 carats) and 0.075 g (0.023 carats). The amount of powder is 0.0045 g and 0.006 g. The paste was applied by spreading evenly over the entire surface of the lapping. Kerosene in an amount of 0.8–0.6 mL was used as coolant. After each experiment, the lapping tool was thoroughly washed and wiped dry. The criteria for evaluating the abrasive material were the finishing performance *Q* mm^3^/min, the roughness of the treated surface, as well as the presence of chips and gouges on the surface and edges of the samples.

Experiments on finishing with a free abrasive were carried out on cast iron lapping tool, grade SCH 28 (cast iron grey), hardness HB 180–220. When finishing on a fixed abrasive, flat circles of 6A2T grades were used as lapping: ASM 28/20–M2-01-2, ASM 40/28–M2-01-2, ASM 60/40–M2-01-2, KM 40/28-M2-01-2.

The main parameters of the processing modes that significantly affect the parameters of the finishing process are the cutting speed and the pressure in the tool –part contact. The cutting speed is determined by the oscillation frequency of the tool and can be adjusted in the range from 0 to 0.253 m/s (15.2 m/min). The creation of the necessary pressure in the tool–treated surface contact was carried out using a pressure device and varied within 5–35 kg (67.7–337 kPa). The experiments were carried out at constant-basic processing modes: nominal pressure *P*_n_ = 270 kPa (2.75 kg/cm^2^), cutting speed *V*_medium_ = 0.212 m/s (12.7 m/min), cycle time of finishing *t*_c_ = 2–4 min.

## 3. Results and Discussion

The basis of the finishing method is a special type of working movement of the tool, which has a complicated trajectory with finely adjustable parameters. The simplest of these trajectories are known as Lissajous figures. Later in the work, the authors use the name of this movement of the tool as a “raster trajectory” [12]. Developed equipment reproducing the movement of the instrument on Lissajous figures-“Raster 220”.

The trajectory of the instrument (Figure 4) represents Lissajous figures, which are formed as a result of an orthogonal combination of two sinusoidal oscillations with different amplitudes *A*, *B* and frequencies *ω*_1_, *ω*_2_. In Cartesian coordinates, this trajectory is described by parametric equations of coordinates *x*, *y* in functions of time *t*: (1)$$x=A\sin\omega_{1}t,\;y=B\sin(\omega_{2}t+\varphi_{0})$$
where *φ*_0_ is the initial phase shift; *ω*_2_ > *ω*_1_, *t* is the time.

Figure 5 shows raster tool trajectories at different ratios of frequencies *ω*_1_, *ω*_2_ and vibration amplitudes of lapp *A*, *B*.

The basic kinematic and geometric parameters of the raster trajectory of the tool movement necessary for practical applications are established. The frame period, the average speed of oscillatory movements of the tool and the grid density of the raster trajectory of the tool are established.

Since the tool moves with equal instantaneous speeds of all points during lapping, the friction paths of the parts are the same across its entire surface. The median cutting speed was determined empirically:$$V_{medium}=3A(n_{1}+n_{2})$$
where *n*_1_ and *n*_2_ are the number of revolutions per minute of the eccentric shafts.

The study of raster trajectories [13] showed that any of them represent a set of consecutive frames, in each of which the curve configuration undergoes a certain cycle of transformations with a frequency equal to the frequency of frame changes, *ω*_f_ = *ω*_2_ − *ω*_1_. The basic geometric and kinematic parameters (speed, acceleration, angle and density of the grid) of the raster trajectory of the lapp working movement are also determined. One of the main parameters determining the nature of the trajectory of the working movement of the tool is the density *q* and the angle *γ* of the grid.

The average distance (or step *P*) between two adjacent lines in the frame characterizes the value of the average grid density *q* of the raster trajectory [12,14]:(2)$$P_{}=2\frac{\omega_{2}-\omega_{1}}{\omega_{2}+\omega_{1}}\sqrt{A^{2}+B^{2}}.$$

The average grid density of the raster trajectory *q* is the number of lines per frame per unit length in the diagonal direction of the frame, and is the average value of the average distance *P*:(3)$$q=\frac{1}{P}=\frac{1}{2\frac{\omega_{2}-\omega_{1}}{\omega_{2}+\omega_{1}}\sqrt{A^{2}+B^{2}}}.$$

The formula for determining the grid density of the raster trajectory, as can be seen, reflects the complex influence of the ratio of frequencies and amplitudes of the folded oscillations.

The angle of inclination of the tangent *tgß* to the motion path determines the direction of the cutting speed vector. The angle change depends on the complexity of the trajectory. The angle of inclination of an arbitrary tangent to the raster trajectory can be written using parametric Equations of motion (1):(4)$$tg\beta =\frac{Y\prime }{X\prime }=\frac{B\omega_{2}\cos(\omega_{2}t+\varphi_{0})}{A\omega_{1}\cos\omega_{1}t}.$$

It is known that during the formation of one frame, a moving point changes the direction of rotation twice. The angle of the grid *γ* be determined by the angle between the tangents to the curve at the beginning and middle of the frame [12,13].

The angle of the tangent at the beginning of the frame: $\beta_{1}=arctg\frac{B\omega_{2}{}_{}}{A\omega_{1}{}_{}}$.

The angle of the tangent in the middle of the frame: $\beta_{2}=-arctg\frac{B\omega_{2}}{A\omega_{1}}$.

Grid angle: $\gamma =\left|\beta_{1}\right|+\left|\beta_{2}\right|$.

The grid angle of the resulting raster trajectory *γ*, as studies have shown [14], practically does not affect the surface roughness, but it affects the finishing performance and the overall texture of the treated surface. When changing the grid angle from 0 to 90°, the productivity increases almost 5 times. Therefore, in order to obtain an anisotropic texture of the microrelief, it is recommended to lapping at equal amplitudes *A*, *B* of the folded oscillations, while the grid angle is close to 90°.

The raster movement overcomes the fundamental disadvantage of the rotational movement of the tool on lapping machines of models–a sharp difference in speeds in the center and on the periphery of the lapp. The raster movement of the tool is inherently progressive, it is characterized by equality of velocities and congruence of trajectories of all points of the cutting surface. This circumstance ensures more uniform wear and full 100% use of the entire tool area during processing.

Thus, it was found that the ratio of the amplitudes of the folded oscillations and frequencies completely determines all the parameters of the raster trajectory of the tool-the angle and density of the grid. This allows you to manage them during processing.

The raster method of abrasive processing of flat surfaces is implemented on an experimental flat-guide machine with a raster trajectory of the “Raster 220” tool.

To solve this problem, studies have been carried out on the precision lapping of the planes of samples from ZrO_2_ ceramics. The influence of various abrasive materials (type, grain size, concentration) and processing modes (processing cycle time, pressure, cutting speed) on the qualitative and quantitative indicators of the process is investigated. All experiments were carried out on a lapping machine “Raster 220”.

The properties of the abrasive materials used in lapping have a significant impact on the processing results. Various abrasive materials in the form of powders and pastes differ in their physical and mechanical properties and performance. The following types of abrasive material were used in the experiments: green silicon carbide powder with a grain size of M28 and M40 microns, a paste of cubic boron nitride (CBN) with a grain size of 28/20 and 40/28 microns and a paste of synthetic diamond (ASM) with a grain size of 7/5, 14/10, 28/20, 40/28 and 60/40 µm.

The experiments were carried out on a lapping made of cast iron of the SCH 28 (cast iron grey) brand with basic processing modes: nominal pressure *P*_n_ = 270 kPa (2.75 kg/cm^2^), the rate of reduction of *V*_midium_ = 0.212 m/s (12.7 m/min), the cycle time of finishing *t*_c_ = 4 min, the amount of paste of CNB and ASM-75 mg, the amount of powder KZ-4.5 mg, coolant-kerosene 0.8–0.6 mg. Each experiment was conducted at least five times, after which the average value was calculated. The results of the experiments are presented in Table 2 and in Figure 6 and Figure 7.

As can be seen from Figure 6, the lowest productivity is observed when using silicon carbide powder (KZ). This is due to the fact that the microhardness of green silicon carbide (27.8–32.3 GPa) is almost commensurate with the microhardness of the ceramics processed by ZrO_2_ (12.5–22 GPa).

Of all the tested abrasive materials, diamond paste ASM showed the highest performance.

The roughness of the treated surfaces does not depend much on the type of abrasive material (Figure 7). However, the least roughness of the treated surfaces is observed when finishing with cubic boron nitride of the CNB, which can be explained by the more rounded shape of the grains compared to the grains of diamond ASM.

Taking into account the high productivity, diamond paste was chosen as the main material for further research.

The results of the study of the process of raster lapping of zirconium ceramic planes with diamond paste of various grain sizes showed that the removal performance and the average roughness value increase proportionally with an increase in the grain size of the paste from ASM 14/10 to ASM 40/28 μm (Figure 8). A slight increase in productivity when processing with ASM 7/5 paste may be due to the fact that the paste grains of this granularity have small radii of rounded tops, which, at the same pressure, allows the grains to penetrate deeper into the processed material and cut off large chips. For the grain size of AFM 60/40, the decrease in productivity can be explained by the larger, compared with 28/20 and 40/28, the size of the radii of the rounded vertices of the grains. In this case, at the same pressure, the force applied to a single grain is not enough to ensure an intensive cutting process (Figure 8).

Table 3 shows photographs of the lapped surface of samples treated with diamond paste ASM of different granularity. It was found that the number of chips and gouges on the edges of the samples increases with increasing grain size, and the deterioration of the quality of the treated surface is also clearly visible.

With increasing grain size, the roughness of the treated surface increases proportionally from *Ra* = 0.015 microns to *Ra* = 0.244 microns (Figure 9).

The influence of the amount of abrasive material applied to the lapp on the parameters of the lapping process and, first of all, on productivity is investigated. The effect of the amount of diamond paste in the range from 75 to 150 mg was studied. The experiments were carried out on a lapp made of cast iron of the SCH 28 brand with basic processing modes. Each experiment was conducted at least five times, after which the average value was calculated. The results of the experiment are presented in Table 4.

During the experiments, it was found that with a decrease in the amount of paste by 2 times, the productivity decreases slightly (Figure 10). Therefore, in order to reduce the consumption of diamonds, it is more expedient to use 75 mg of paste. The roughness of the treated surface practically does not depend on the amount of paste and varies from *Ra* = 0.1322 µm to *Ra* = 0.1800 µm (Figure 11).

With an increase in the amount of paste, the finishing performance increases slightly. The roughness of the treated surface practically does not depend on the increase in the amount of paste from 75 to 150 mg. When fine-tuning ceramic samples to save paste, the optimal amount applied to the surface of the paste with a diameter of 220 mm paste is 75 mg.

The processing time is most often determined empirically, since it depends on a large number of factors: the lapping material and the parts being processed, the material and consistency of the abrasive mixture, processing modes and the amount of allowance for finishing. To determine the optimal time of the ceramic processing cycle, experiments were carried out on a single “spread”. Every 1–2 min, the productivity and roughness of the treated surface were measured. Table 5 shows the results of the experiment.

It follows from the results of the experiment that with an increase in the lapping time from 1 to 6 min, productivity does not change significantly, further increase in processing time is not advisable, since productivity decreases sharply (Figure 12). This is due to the fact that after 6 min of processing on one spread, the abrasive mixture (diamond paste + COOLANT) quickly loses its cutting properties due to grinding, blunting of abrasive grains and accumulation of wear products (sludge). Therefore, the optimal processing cycle time is 2–4 min.

As the lapping time increases, the roughness of the treated surfaces decreases from *Ra* 0.1807 µm to *Ra* 0.1322 µm, which is associated with the gradual grinding of abrasive grains (Figure 13).

The cutting speed during raster lapping of planes is an important parameter, which is determined by the frequency of lapp vibrations. During the research, the cutting speed varied in the range of *V*_midium_ = 0.071–0.212 m/s (4.26–12.7 m/min). Each experiment was repeated at least five times, after which the average value was calculated. The results of the experiment are presented in Table 6.

An increase in the lapp cutting speed is accompanied by an increase in productivity. An increase in productivity is observed with an increase in the cutting speed from 0.106 (6.36 m/min) to 0.141 m/s (8.46 m/min), a further increase in the cutting speed to 0.212 m/s (12.72 m/min) has little effect on productivity growth (Figure 14). This can be explained by more intensive thickening and drying of the abrasive mixture (diamond paste + COOLANT) at high speeds of lapp and possible more intensive grinding of paste grains.

As the cutting speed increases, the roughness of the treated surface changes slightly from *Ra* = 0.1739 µm to *Ra* = 0.1587 µm (Figure 15).

The conducted studies have shown that it is more expedient to lapping ZrO_2_ ceramics at a cutting speed no higher than 0.212 m/s (12.72 m/min).

The pressure of the parts on the lapp is transmitted through the diamond grains located between them. Obviously, the pressure determines the amount of load on each working grain and affects the conditions of micro-cutting. During the experiments, the nominal pressure varied in the *P*_n_ range = 67.7–337 kPa (0.7–3.4 kg/cm^2^). Each experiment was conducted at least five times, after which the average value was calculated. The results of the experiment are presented in Table 7.

From Table 7 and Figure 16, it can be seen that productivity increases with an increase in pressure from 67.7 (0.68 kg/cm^2^) to 270 kPa (2.8 kg/cm^2^) and changes little with a further increase to 337 kPa (3.4 kg/cm^2^). The latter circumstance can obviously be explained by grinding diamond paste grains under the influence of a heavy load. It should be noted that when lapping with a pressure above 337 kPa (3.4 kg/cm^2^), cracks and gouges begin to appear on the surface of the samples (Figure 17), which is not permissible.

As the pressure increases, the roughness of the treated surface changes slightly in the range from *Ra* 0.1539 µm to *Ra* 0.1633 µm (Figure 18).

Thus, it was found that the optimal pressure value for processing samples from ZrO_2_ ceramics should be considered 203–270 kPa (2.1–2.8 kg/cm^2^).

The method of lapping with a fixed abrasive has been widely used with the advent of synthetic diamonds and other superhard abrasive materials in the finishing of products made of technical ceramics. The main parameters of the characteristics of a tool with a fixed abrasive are the grade and grain size of the abrasive, the type of bundle and the concentration of the abrasive in the bundle. The most widely used tool is made of synthetic diamonds on metal bundles.

Compared with free abrasive processing, the process of lapping with a fixed abrasive has a number of significant advantages: it does not require a large consumption of abrasive material, less labor-intensive for washing operations. Therefore, experimental testing was carried out in order to determine the possibility of using this method for processing ZrO_2_ ceramics. The experiments were carried out according to the same technique as finishing with a free abrasive, on diamond circles of the 6A2T shape, on a metal bundle M2-01, with a concentration of 2 (50%), of different grain sizes 60/40, 40/28 and 28/20 microns. Comparative experiments on lapping with free and fixed abrasives were carried out under identical conditions, the results of the study are presented in Table 8.

The experimental data shown in Table 8 show that the highest productivity (0.16 mm^3^/min) on diamond lapping with a grain size of AFM 40/28 and AFM 28/20 microns is lower than the processing performance on cast iron lapping paste with a grain size of AFM 14/10 microns. This can be explained by the fact that with a high hardness of ZrO_2_ ceramics and a relatively low pressure of up to 270 kPa, the entire load is transferred to the tops of the diamond grains. However, their introduction into the treated surface is very insignificant, which is not enough to implement an effective cutting process. The material consumption is negligible.

Figure 19 shows a graph of the dependence of processing performance on the granularity of diamond lapp.

With an increase in grain size, as well as in the case of processing with a free abrasive, there is a sharp decrease in productivity. Grain sizes and the radii of rounding of their vertices at the diamond lapping ASM 60/40 microns when processed with the maximum allowable pressure, do not allow creating sufficient force applied to a single grain to ensure the cutting process.

Lapping on diamond wheels is significantly inferior in performance to finishing with a free abrasive of the same grain size, but differs in a lower and stable roughness of the treated surface (Figure 20). With a decrease in grain size from ASM 60/40 M2-01-2 to ASM 28-20 M2-01-2, the roughness decreases from *Ra* = 0.0694 µm to *Ra* = 0.0495 µm.

In addition, the quality of the lapped surfaces was assessed by the presence of chips and gouges, which were recorded by photographing the edges of the samples after processing with a microscope (Table 9). As a result of visual analysis of the surface, it can be concluded that the number of chips and gouges on the edges of the samples increases with increasing granularity of diamond grains.

Compared with free abrasive processing, the process of lapping with a fixed abrasive has a number of significant advantages: it does not require a large consumption of abrasive material, is less labor-intensive for washing operations, and also has a lower and stable roughness of the treated surface. However, lapping on a fixed abrasive is significantly inferior in performance to finishing with a free abrasive under the same conditions. Therefore, it is impractical to use this method for processing ZrO_2_ ceramics.

The distribution of the machining forces between individual abrasive grains is very nonuniform. For example, in the finishing of plates, the cutting force is considerably greater at points of nonplanarity of the surface. In other words, abrasive finishing is very sensitive to change in shape of the surface. In abrasive finishing, the problem is that change in some parameter of the process will change the interaction of a whole set of factors. Consequently, some assumptions in shaping calculations cannot be applied to the whole set of practical problems relating to the abrasive finishing of surfaces.

On the basis of the Preston hypothesis, which states that, in cutting, the thickness of the layer of material removed is proportional to the energy consumed in its removal, we may write a dynamic relation for abrasive finishing:$$h=\int_{0}^{t}AvPdt$$

Here *h* is the height of the layer removed in abrasive finishing; *P* is the contact pressure; *v* is the mean cutting speed; and the technological constant *A* = *V*/*P*_*ra*_, where *V* is the mean rate of removal of the material; and *P*_*ra*_ is the rated pressure.

In physical terms, *A* is the intensity of wear. Usually, in finishing, the dependence of *V* on *P*_*ra*_ and *v* is practically linear. Therefore, *A* does not depend on the pressure and the rate of removal of the material [15].

Initially, in running-in, there are only a few contact spots between surfaces. These spots form the contact area, at which the contact pressure will considerably exceed *P*_*ra*_. Derived the dependence of the mean contact pressure *P* on the shape deviation δ by means of a solution of the Hertz problem based on elasticity theory [16].

In the formation of the geometric shape, the excess of the total contact pressure above *P*_*ra*_ plays a regulatory role, since increase in pressure at a convex section leads to local increase in the material removed. Thus, the dependence of *P* on *δ* reflects both the monitoring and regulatory elements of the control system. The coefficient *K* in this dependence is the rate of change in the contact pressure. The dependence *P*(*δ*) may be linearized if we assume that *K* is constant and equal to its mean value. In that case, it is simply a constant of proportionality: *P = K**δ*.

If we regard *v* as the mean relative velocity and assume constant friction then:(5)$$h=Av\int_{0}^{t}Pdt$$

In the course of running-in, as wear proceeds, the shape deviation *δ* declines from its initial value *δ*_0_,
(6)$$\delta =\delta_{0}-h$$

The excess pressure at the contact spot will change accordingly:(7)$$P=K(\delta_{0}-h)$$

Differentiating both sides of Equation (7) with respect to the time, we obtain,
$$\frac{dh}{dt}=\frac{dAv_{me}\int_{0}^{t}Pdt}{dx}$$

After the substitution of Equation (7), we obtain a uniform first-order differential equation with separable variables,
$$\frac{dh}{dt}=Av_{me}(K(\delta_{0}-h))$$

After grouping the variables and differentials, we obtain,
$$\frac{dh}{\delta_{0}-h}=Av_{me}Kdt$$

Integration yields,
$$\int \frac{dh-\delta_{0}}{\delta_{0}-h}=\int -Av_{me}Kdt$$
$$\ln(h-\delta_{0})=-Av_{me}Kt+C\;$$

Hence, we find that,
$$h-\delta_{0}=e^{-Av_{me}Kt+C}$$

The general solution takes the form,
(8)$$h=\delta_{0}+e^{C}e^{-Av_{me}Kt}$$

With the initial condition that *h* = 0 when *t* = 0, we find that,
$$\delta_{0}+e^{C}e^{-Av_{me}K0}=0$$

Then *e^C^* = −*δ*_0_. Substituting *e^C^* into Equation (8), we obtain,
$$h=\delta_{0}-\delta_{0}e^{-Av_{me}Kt}$$

The final solution takes the form,
(9)$$h=\delta_{0}(1-e^{-Av_{me}Kt})$$

Taking account of Equation (2), we obtain an equation for the flattening out of the microrelief in the course of running-in,
$$\delta =\delta_{0}e^{-\frac{t}{T_{sh}}}$$

In physical terms, the time constant *T*sh = (*AvK*)^−1^ is the time for the geometric deviation to fall to zero at a constant rate equal to the initial rate. In practice, the duration *t*tr of the transient process is at least 3*T*_sh_. (Here *δ <* 0.05 *δ*_0_.) In steady conditions, the system operates without residual deviation: that is, *δ* = 0 as *t* → ∞. Such control systems are said to be astatic.

The variation in contact area during the transient process is of most interest. In the approximation of a geometric relation between the diameter d of the contact spot, the radius of curvature R of the macroscopic nonuniformity, and the thickness h of the layer removed, we may write,
$$d^{2}=8Rh$$

At an arbitrary time, the contour area *S* may be written in the form,
$$S=\frac{\pi d^{2}}{4}\;\mathrm{or}\;S=2\pi Rh$$

In particular, the substitution *h =*
*δ*_0_ yields the rated contact area,
$$S_{ra}=2\pi R\delta_{0}$$

With constant overall load, the contour area *S* of the contact spot tends to the rated value *S*ra. Therefore, we may write,
$$\frac{S}{S_{ra}}=\frac{h}{\delta_{0}}\;\mathrm{or}\;S=S_{ra}\left(\frac{h}{\delta_{0}}\right)$$

Substituting *h* from Equation (5), we obtain the time dependence of the contour area,
$$S=S_{ra}\left(1-e^{{}^{-\frac{t}{T_{sh}}}}\right)$$

The transient process ends when the contact pressure reaches the rated value. Then, the formation of the surface is a steady process, with constant wear rate *V* = *AvP*_*ra*_. The total linear surface wear *H* consists of *h* and *Vt*_st_,
$$H=\delta_{0}(1-e^{-Av_{me}Kt})+AVP_{ra}t_{st}\;$$

On the basis of the foregoing,
$$F=F_{0}+F_{ra}\left(1-e^{{}^{-\frac{t}{T_{sh}}}}\right)$$
where *F* is the clamping force and *F*_0_ is the initial clamping force. Thus, if we take account of the variation in part–tool contact area and control the clamping force on the lap in proportion to the contact area, we may theoretically reduce the machining time by a factor of 2–3, with more stable pressure in the machining zone. In addition, these conditions are better for brittle and nonrigid materials, since the rated forces recommended in the literature are not always employed.

Experiments on the abrasive finishing of CuSn5Zn5Pb5-C zinc samples (State Standard GOST 613–79) permit verification of the Preston hypothesis. The requirements for the accuracy of the geometric shape of the lapped samples are a deviation from the flatness of no more than 1 µm, the roughness of the finished surface according to the parameter Ra is no more than 0.1 µm. The cutting tool employed in these experiments is bound boron-nitride abrasive on 6A2T plane wheels (State Standard GOST 17007–80), with M2-01 metallic binder. The granularity of the abrasive is 40/28 μm, and its concentration score is 2 (50%).

The equipment employed is the “Rastr 220” planefinishing machine with a raster tool trajectory [6,12].

The experiments are conducted with constant (6.4 kg) and variable (0–34 kg) force *F* clamping the part to the cutting tool. In all the experiments, the mean cutting speed is 15 m/min. In the abrasive process, we are interested in the formation of macro- and microrelief and the productivity. The microrelief of the machined surface is estimated by measuring the nonlinearity of the surface in the radial direction. In Figure 21, we show the time characteristics of the transient process. The Table 10 illustrates the variation in contour area of the workpiece in the course of machining.

After each experiment, the machined surface is scanned and then the image is processed by means of specialized software, to determine the contact area. We note that material is removed considerably more rapidly from parts machined with variable clamping force (Table 10).

On the basis of the nonlinearity data, we conclude that the nonlinearity of the machined surface is practically the same in the cases of constant and variable clamping force. However, the time to attain minimal nonlinearity is smaller by a factor of three in the case of variable clamping force (Table 10).

We also find that the rate of removal of the material is higher with variable clamping force. This may be explained in that, in the case of variable clamping force, the losses in the distribution of the contact pressure over the contact area are compensated with exponential increase in the load. We show the linear removal of material with constant and variable clamping force in Figure 22a,b, respectively.

The microrelief of the machined surfaces is estimated on a Mahr S2 profilometer. From the results, we calculate the mean square deviation *σ* of the roughness *Ra* [17]. Studies have found that when processing with variable clamping force, a 35–40% reduction in the standard deviation of *σ* parameters of surface roughness is provided. This can be explained by the constant pressure on each single grain over time and a more stable micro-cutting process.

With exponential variation in the force-that is, variation analogous to that of the contact area in abrasive finishing-the productivity is increased by a factor of 2.5–3. Our results permit numerical and visual assessment of the shaping of plane surfaces in abrasive finishing (lapping) with constant and variable clamping force. The proposed loading method permits effective machining of brittle and nonrigid parts.

## 4. Conclusions

The regularities of the formation of the resulting raster trajectories based on Lissajous figures for the process of lapping flats are established. The possibility of controlling the main characteristics (configuration, density and angle of the grid of processing traces) by changing the values and ratios of the parameters of the elementary movements of the tool is shown. This makes it possible to maximize the cutting ability of the tool, which contributes to its more uniform wear and increased productivity and processing quality. Based on the Preston hypothesis, an exponential dependence of the change in the contact area during the fine-tuning of planes of different initial shape of the macrorelief is obtained. The obtained theoretical and practical results of the study of the process of flat lapping with constant and variable clamping force of the treated surface to the surface of the tool. Based on the results of the study, the following conclusions and recommendations can be made.

An experimental test of Preston’s assumption and hypothesis was carried out at constant (6.4 kg) and variable clamping force, which showed that the time to achieve the smallest deviation from straightness with variable clamping force is three times less than with constant clamping force.The change in the standard deviation σ of the roughness parameter Ra of the treated surface during lapping with a constant clamping force at time t = 1 min is almost twice as high as the standard deviation of the roughness at the same time with a variable clamping force.During the raster lapping of zirconium (ZrO_2_) ceramics, the diamond paste ASM showed the greatest performance of all the tested abrasive materials, however, the surface roughness of the diamond paste brought is somewhat coarser than the powder made of silicon carbide green (KZ) and paste made of cubic boron nitride (CNB). Considering all of the above, it is more expedient to use diamond synthetic paste when lapping high-hardness zirconium ceramics.The graininess of the paste has a significant impact on the lapping performance and surface roughness. A larger (60/40 µm) grain size is ineffective, since it does not provide an increase in productivity. Optimal parameters of productivity and roughness of the treated surface during lapping of zirconium ceramics are achieved through the use of ASM paste 28/20 µm.With an increase in the amount of diamond synthetic paste, the lapping performance does not increase significantly. The roughness of the treated surface practically does not depend on the increase in the amount of paste from 75 to 150 mg.With an increase in the finishing time from 1 to 6 min on one spread, productivity changes slightly, further increase in processing time is not advisable, since productivity decreases. Therefore, the optimal cycle time for processing ZrO_2_ ceramics on one spread is 2–4 min.An increase in the cutting speed from 0.106 (6.36 m/min) to 0.141 m/s (8.46 m/min) is accompanied by a noticeable increase in productivity, a further increase in the cutting speed to 0.212 m/s (12.72 m/min) has little effect on the increase in removal. The roughness of the treated surface practically does not depend on the change in cutting speed.With increasing pressure, productivity increases, the roughness of the treated surface changes slightly. When lapping with a pressure above 337 kPa (3.4 kg/cm^2^), cracks and gouges begin to appear on the surface of the samples, which is not permissible. The optimal pressure value when processing ZrO_2_ ceramics should be considered 203–270 kPa (2.1–2.8 kg/cm^2^).In comparison with the processing with a free abrasive, the lapping process with a fixed abrasive has a number of significant advantages: it does not require a large consumption of abrasive material, is less labor-intensive for washing operations, and also has a lower and stable roughness of the treated surface. However, lapping on a fixed abrasive is significantly inferior in performance to lapping with a free abrasive under the same conditions.

## Figures and Tables

**Figure 1 materials-15-03048-f001:**
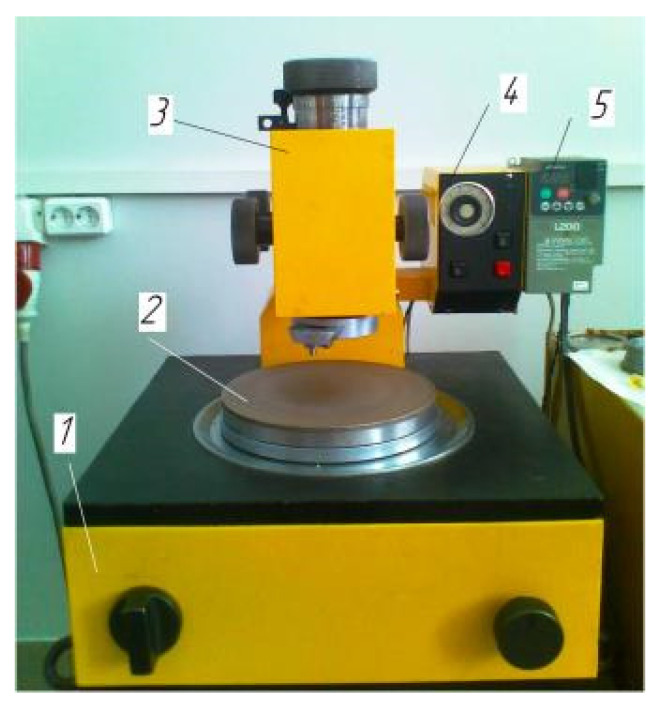
Lapping machine “Raster 220”. 1—drive unit, 2—Lapp, 3—pressure device, 4—control panel, 5—frequency converter.

**Figure 2 materials-15-03048-f002:**
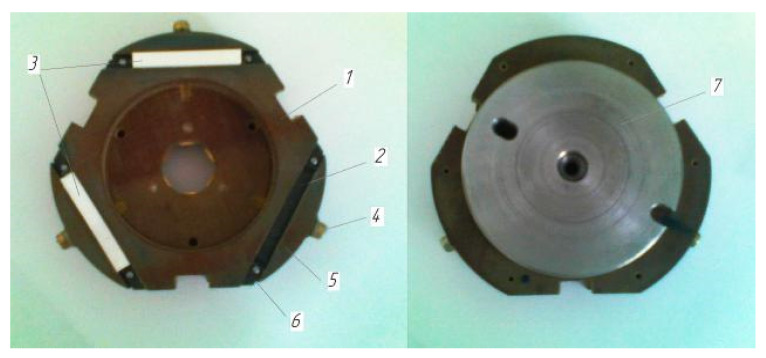
Device for processing ceramic samples. 1—body, 2—grooves, 3—machined parts, 4—screw, 5—clamp, 6—elastic limiters, 7—washer.

**Figure 3 materials-15-03048-f003:**
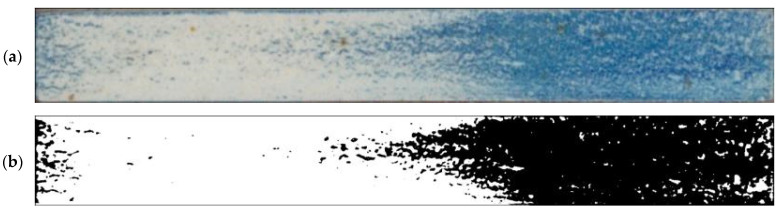
Scanned image of the finished surface of the parts; (**a**) the original, (**b**) processed in the program.

**Figure 4 materials-15-03048-f004:**
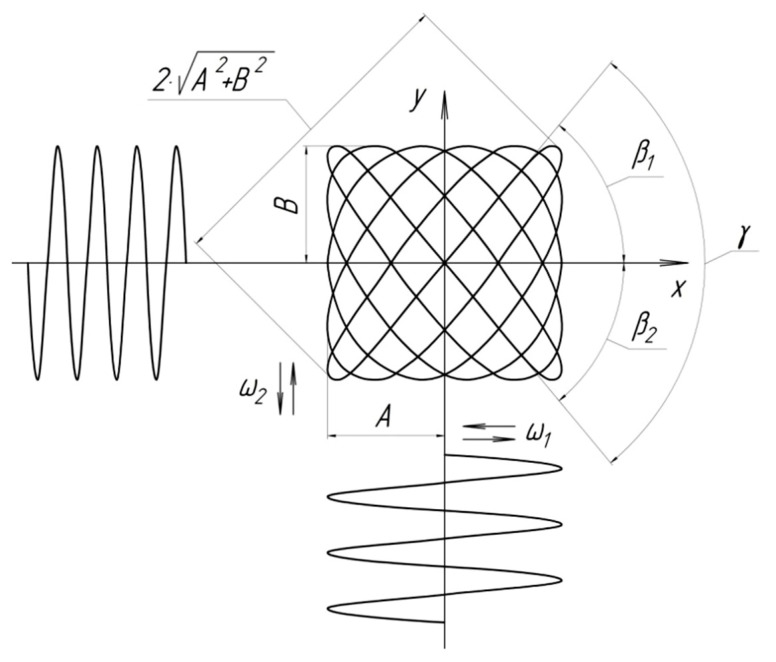
The trajectory of the working movement of the tool when lapping planes.

**Figure 5 materials-15-03048-f005:**
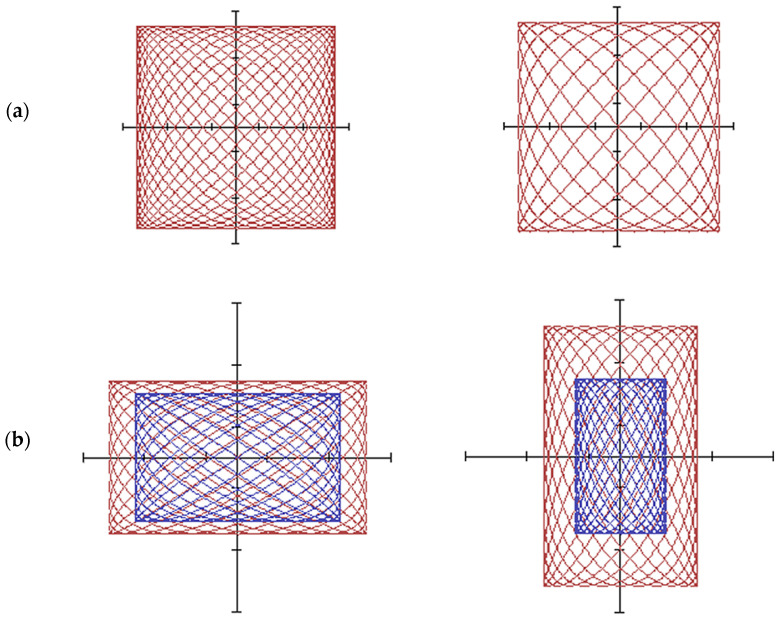
The trajectory of the tool: (**a**) at different ratios of frequencies *ω*_1_, *ω*_2_; (**b**) at different oscillation amplitudes *A* and *B*; (**c**) at different ratios of equal oscillation amplitudes.

**Figure 6 materials-15-03048-f006:**
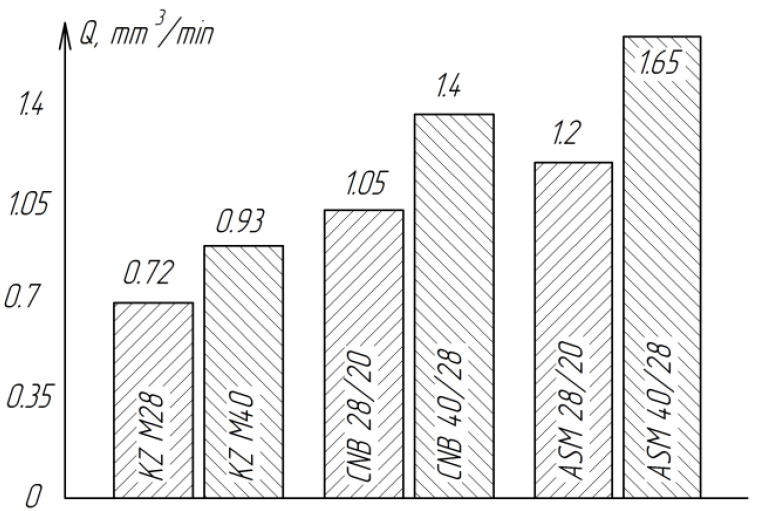
Influence of the type and grain size of the abrasive material on the processing performance of ZrO_2_ ceramics.

**Figure 7 materials-15-03048-f007:**
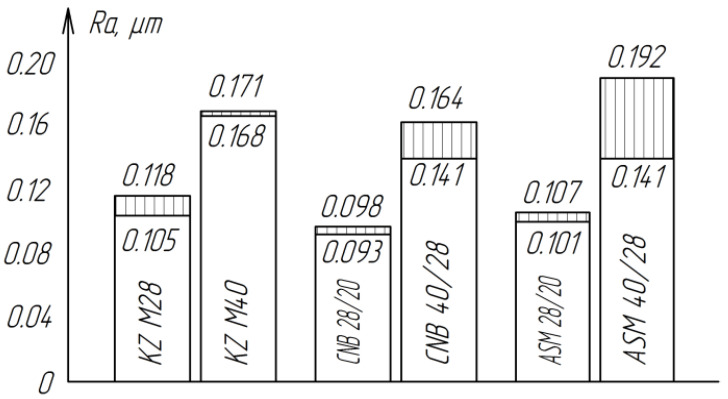
Influence of the type and grain size of the abrasive material on the roughness of the treated surface of ZrO_2_ ceramics.

**Figure 8 materials-15-03048-f008:**
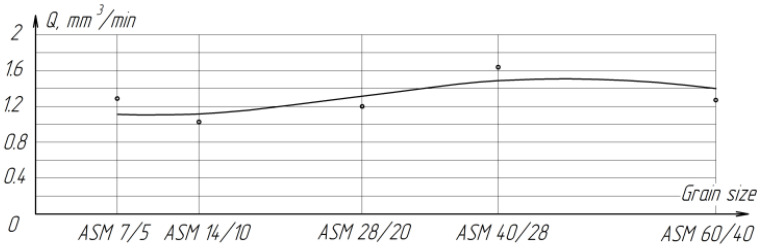
The effect of grain size on the processing performance of ZrO_2_ ceramics.

**Figure 9 materials-15-03048-f009:**
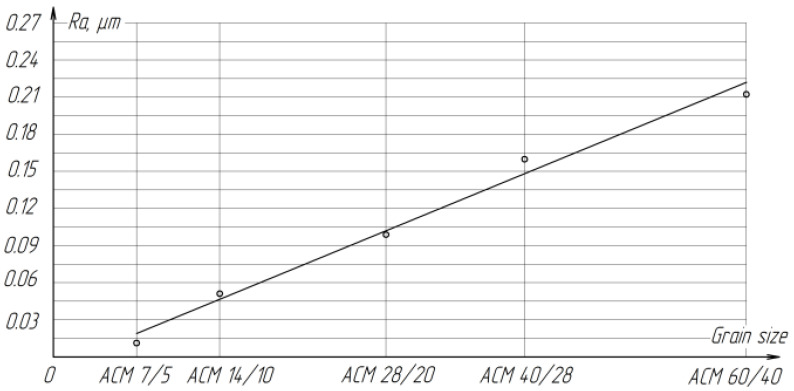
The effect of grain size on the roughness of the treated surface of ZrO_2_ ceramics.

**Figure 10 materials-15-03048-f010:**
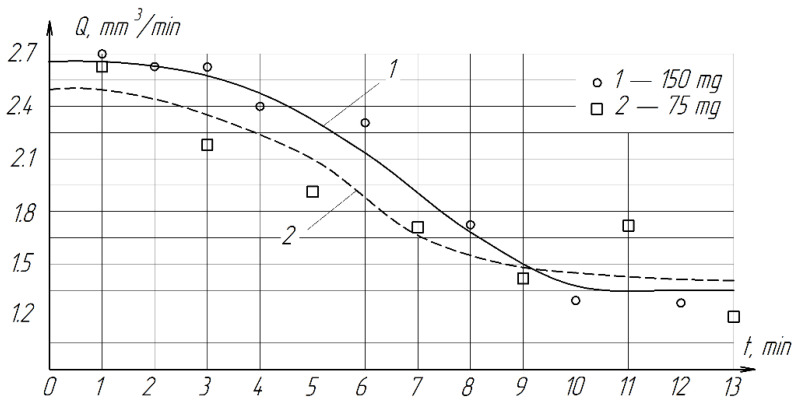
The effect of the amount of diamond paste on the processing performance of ZrO_2_ ceramics.

**Figure 11 materials-15-03048-f011:**
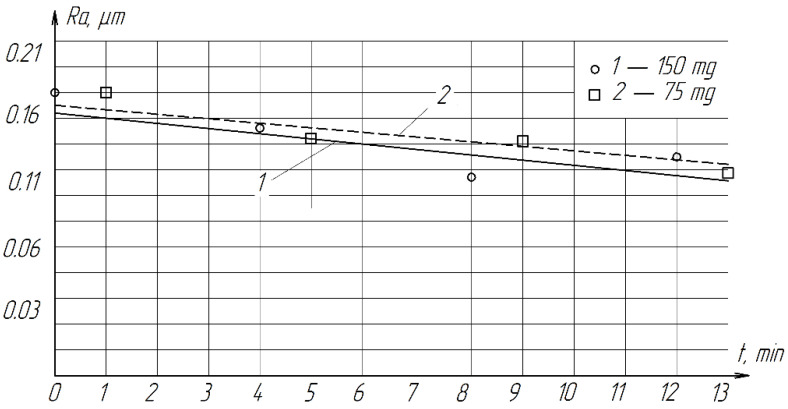
The effect of the amount of diamond paste on the roughness of the treated surface of ZrO_2_ ceramics.

**Figure 12 materials-15-03048-f012:**
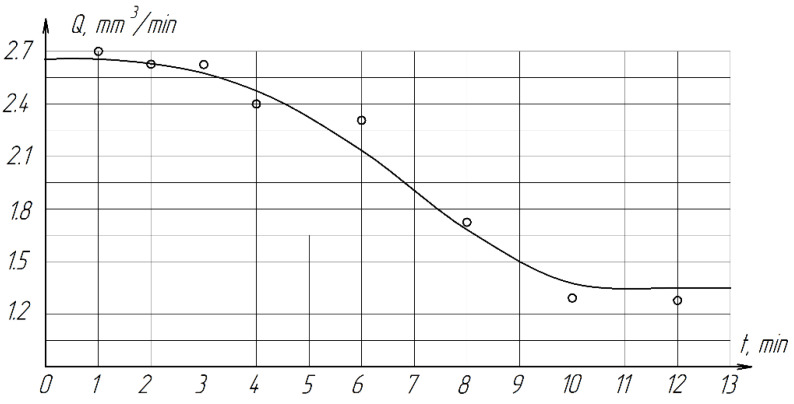
The effect of the total processing time T on productivity.

**Figure 13 materials-15-03048-f013:**
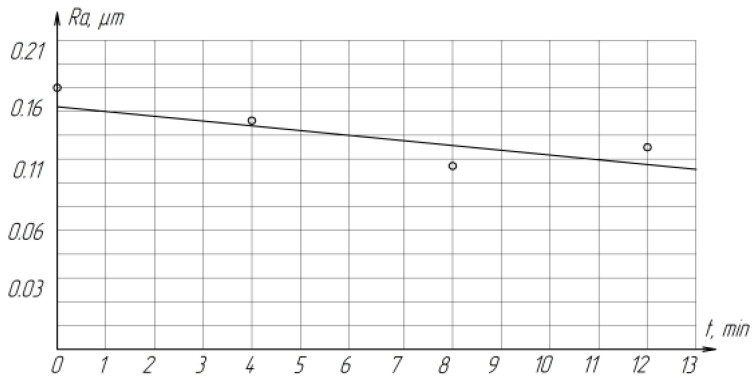
The effect of the total processing time T on the roughness of the treated surface of ZrO_2_ ceramics.

**Figure 14 materials-15-03048-f014:**
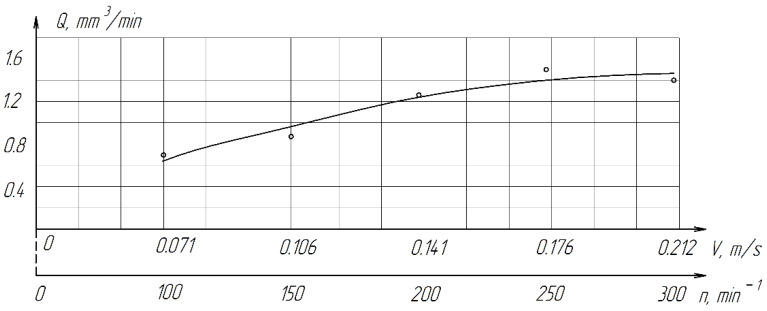
The effect of cutting speed on the processing performance of ZrO_2_ ceramics.

**Figure 15 materials-15-03048-f015:**
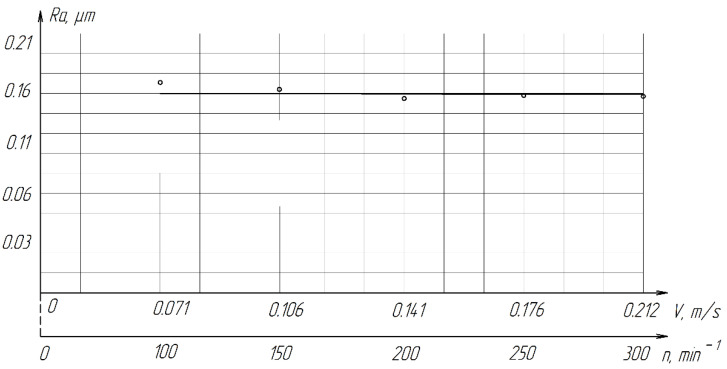
The effect of the cutting speed on the roughness of the treated surface of ZrO_2_ ceramics.

**Figure 16 materials-15-03048-f016:**
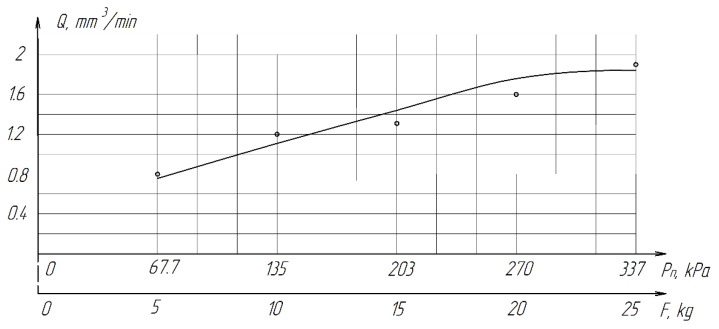
The effect of *P*_n_ pressure on the processing performance of ZrO_2_ ceramics.

**Figure 17 materials-15-03048-f017:**
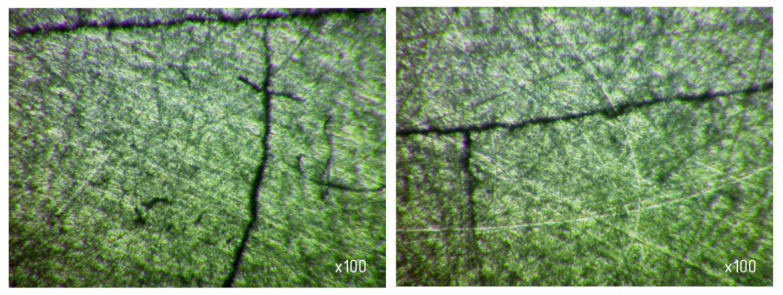
Cracks and gouges on the treated surface after lapping with a pressure of 337 kPa (3.4 kg/cm^2^) (magnification ×100).

**Figure 18 materials-15-03048-f018:**
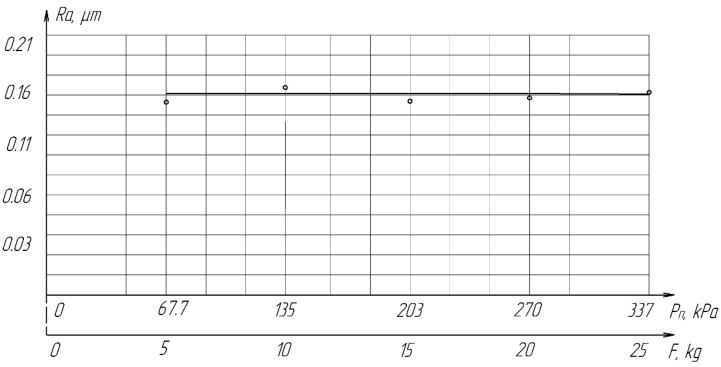
The effect of *P*_n_ pressure on the roughness of the treated surface of ZrO_2_ ceramics.

**Figure 19 materials-15-03048-f019:**
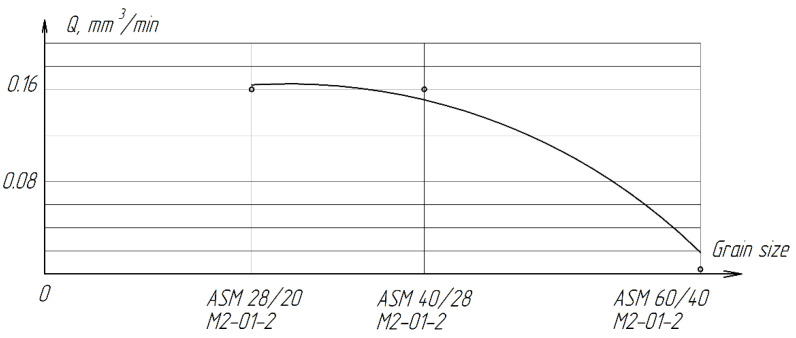
The effect of grain size on the processing performance of ZrO_2_ ceramics.

**Figure 20 materials-15-03048-f020:**
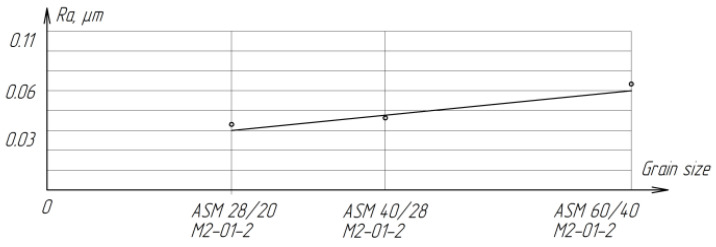
The effect of grain size on the roughness of ZrO_2_ ceramics.

**Figure 21 materials-15-03048-f021:**
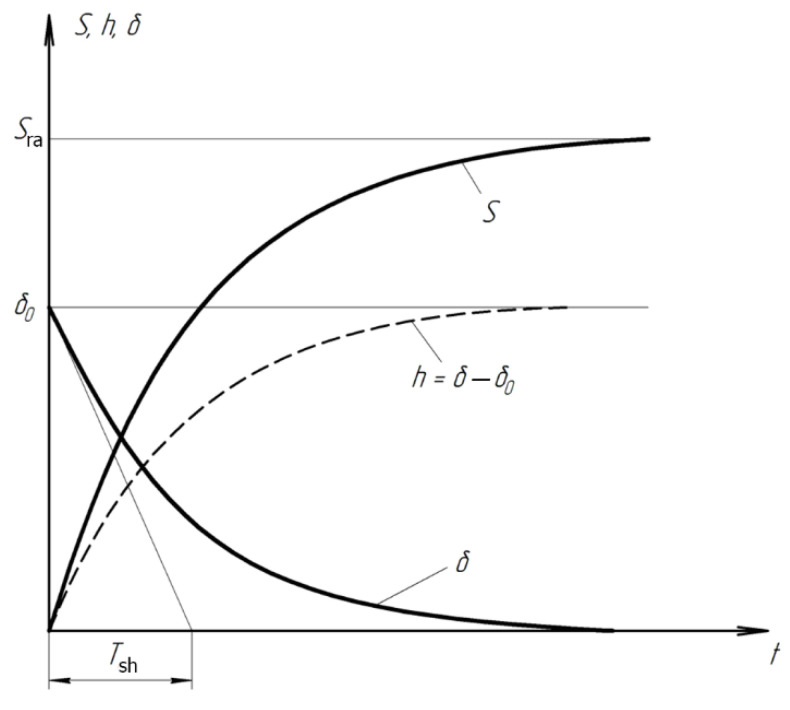
Time characteristics of the transient finishing process.

**Figure 22 materials-15-03048-f022:**
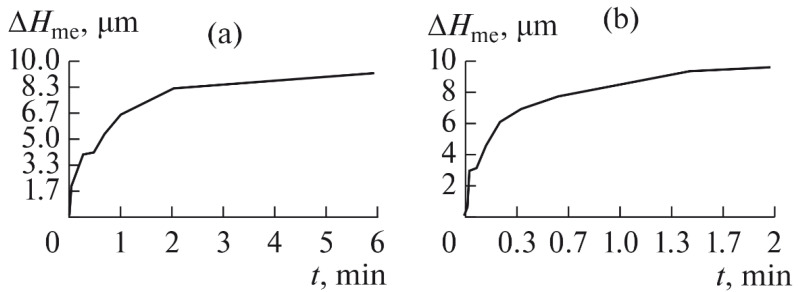
Linear removal Δ*H*me of the material with the same constant (6.4 kg) (**a**) and variable (0–34 kg) (**b**) clamping forces.

**Table 1 materials-15-03048-t001:** Basic numerical data of the machine.

Diameter of the lapp, mm	220
Trajectory frame dimensions, mm	14 × 14
Vibration frequency of the faceplate, 1/min	0–360
Detuning of oscillation frequencies, %	0–15
Average speed of the lapp, m/s (m/min)	0–0.253 (0–15.2)
Clamping force of parts, kgf	0–40
Drive power, kW	0.18
Overall dimensions, mm	400 × 500 × 520
Machine weight, kg	120

**Table 2 materials-15-03048-t002:** Influence of the type and grain size of the abrasive material on the productivity and roughness of the lapped surface.

Abrasive	*Q*, mm^3^/min	*Ra*_midium_, µm	Conditions
KZ M28 (4.5 mg)	0.72	0.1112	Cast iron lapp,
CNB 28/20 (75 mg)	1.05	0.0955	*P*_n_ = 270 kPa,
ASM 28/20 (75 mg)	1.2	0.1039	*V*_midium_ = 0.212 m/s,
KZM40 (6 mg)	0.93	0.1692	*T* = 4 min,
CNB 40/28 (75 mg)	1.4	0.1526	Coolant: kerosene 0.8–0.6 mL
ASM 40/28 (75 mg)	1.65	0.1661	

**Table 3 materials-15-03048-t003:** Photos of the surface finished with diamond paste of different grain size (magnification ×100).

Grain Size, μm	Surface after Processing
ASM 7/5	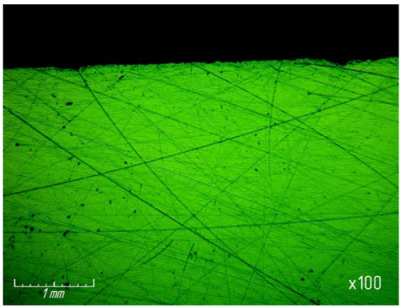
ASM 14/10	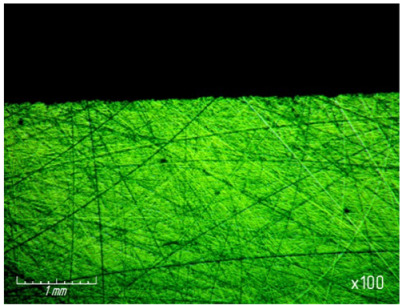
ASM 28/20	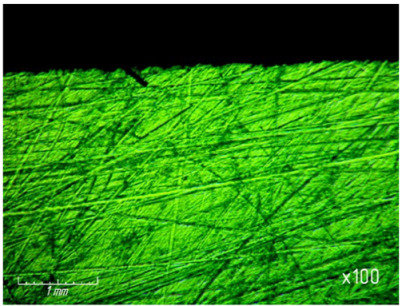
ASM 40/28	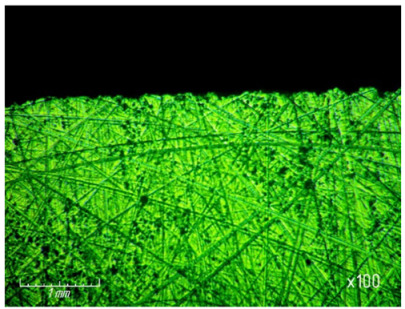
ASM 60/40	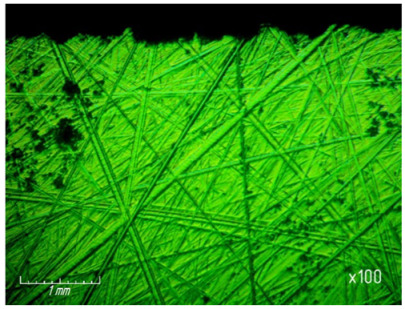

**Table 4 materials-15-03048-t004:** The effect of the amount of diamond paste on the productivity and roughness of the treated surface.

Quantity of Paste	*Q*, mm^3^/min	*Ra*_midium_, µm	Conditions
150 mg	2.6	0.1563	Cast iron lapp, paste ASM 28/20, *P*_n_ = 270 kPa, *V*_midium_ = 0.212 m/s, Coolant: kerosene 0.8–0.6 mL
75 mg	2.1	0.1472	

**Table 5 materials-15-03048-t005:** The effect of processing time on the productivity and roughness of the lapped surface.

Total Processing Time *t*, min	*Q*, mm^3^/min	*Ra*_midium_, µm	Conditions
initial		0.1807	Cast iron lapp, paste ASM 28/20, paste weight 75 mg *P*_n_ = 270 kPa, *V*_midium_ = 0.212 m/s, Coolant: kerosene 0.8–0.6 mL
1	2.7	0.1563	
2	2.6		
3	2.6		
4	2.4		
6	2.3	0.1181	
8	1.7		
10	1.3	0.1322	
12	1.3		

**Table 6 materials-15-03048-t006:** The effect of the cutting speed (frequency of lapping vibrations) on the processing performance and roughness of the lapped surface.

*V*, m/s (m/min)	*n*, min^−1^	*Q*, mm^3^/min	*Ra*_midium_, µm	Conditions
0.071 (4,26)	100	0.7	0.1739	Cast iron lapp, paste ASM 28/20, paste weight 75 mg *P*_n_ = 270 kPa, *t* = 4 min, Coolant: kerosene 0.8–0.6 mL
0.106 (6.36)	150	0.89	0.1670	
0.141 (8.46)	200	1.25	0.1585	
0.176 (10.56)	250	1.5	0.1594	
0.212 (12.72)	300	1.4	0.1587	

**Table 7 materials-15-03048-t007:** Influence of pressure on the productivity and roughness of the lapped surface.

Clamping Force *F*, kg	*P*_n_, kPa (kg/cm^2^)	*Q*, mm^3^/min	*Ra*_midium_, µm	Conditions
5	67.7 (0.68)	0.8	0.1539	Cast iron lapp, paste ASM 28/20, paste weight 75 mg *V*_midium_ = 0.212 m/s, *t* = 4 min, Coolant: kerosene 0.8–0.6 mL
10	135 (1.4)	1.2	0.1677	
15	203 (2.1)	1.3	0.1587	
20	270 (2.8)	1.6	0.1597	
25	337 (3.4)	1.9	0.1633	

**Table 8 materials-15-03048-t008:** Results of lapping ZrO_2_ ceramics with free and fixed abrasive.

Characteristics of the Abrasive Layer	*Q*, mm^3^/min	*Ra*_midium_, µm	Conditions
Processing with a Fixed Abrasive			
Lapp ASM 60/40 M2-01-2	0.013	0.0689	*P*_n_ = 270 kPa, *V*_midium_ = 0.212 m/s, *t* = 4 min, Coolant: kerosene 0.8–0.6 mL
Lapp ASM 40/28 M2-01-2	0.16	0.0492	
Lapp ASM 28/20 M2-01-2	0.16	0.0463	
Free Abrasive Processing			
Paste ASM 40/28	1.63	0.1661	Cast iron lapp, paste ASM 28/20, paste weight 75 mg, *P*_n_ = 270 kPa, *V*_midium_ = 0.212 m/s, Coolant: kerosene 0.8–0.6 mL
Paste ASM 28/20	1.2	0.1039	
Paste ASM 14/10	1.05	0.0519	

**Table 9 materials-15-03048-t009:** Photos of the surface after processing on different grain sizes.

Grain Size, μm	Surface after Processing
ASM 28/20	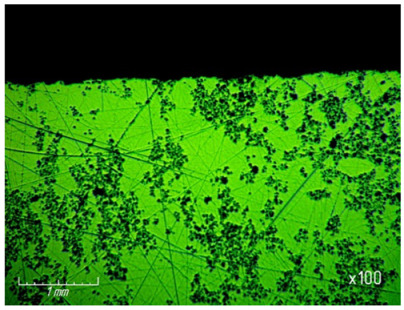
ASM 40/28	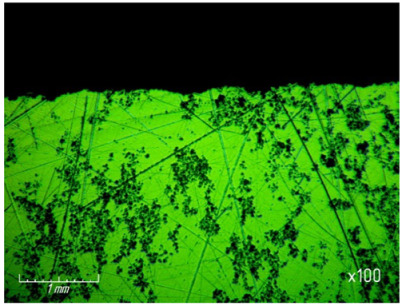
ASM 60/40	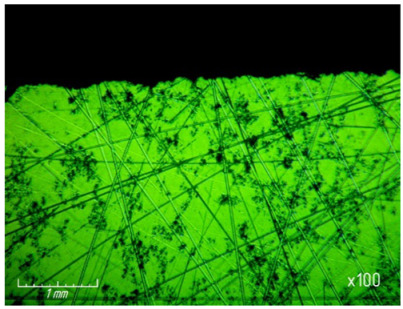

**Table 10 materials-15-03048-t010:** Variation in contour area of the machined surface.

Photographs of Machined Surface				
*S* = 43 mm^2^	*S* = 672 mm^2^	*S* = 849 mm^2^	*S* = 1017 mm^2^	*S* = 1141 mm^2^
**Lapping with Constant Clamping Force (6.4 kg)**				
t = 0 min	t = 0.17 min	t = 1 min	t = 2 min	t = 6 min
Δ = 10.88 µm	Δ = 6.40 µm	Δ = 2.55 µm	Δ = 1.29 µm	Δ = 0.48 µm
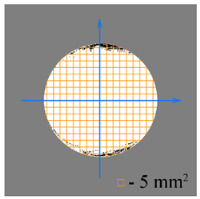	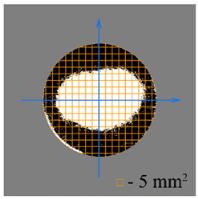	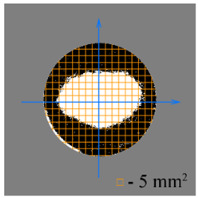	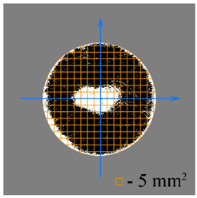	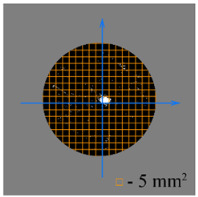
**Lapping with variable clamping force (0–34 kg)**				
t = 0 min	t = 0.17 min	t = 0.5 min	t = 1 min	t = 2 min
Δ = 10.17 µm	Δ = 4.82 µm	Δ = 2.87 µm	Δ = 1.29 µm	Δ = 0.55 µm
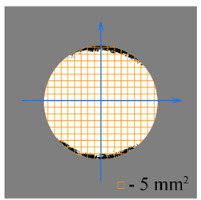	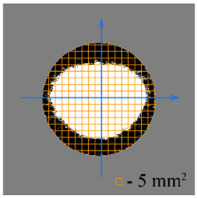	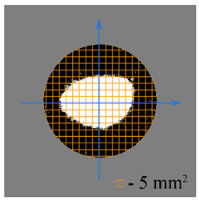	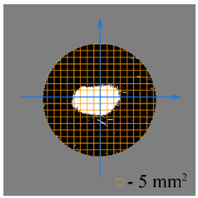	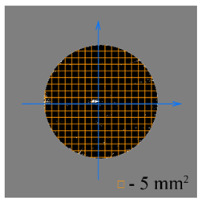

## Data Availability

The authors have all the data and materials of the study.

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
