# Peer review of "Investigation of Technological and Load Intensity Parameters of the Finishing Process of Materials on Equipment with Tools Translational Kinematics"

_materials, 2022, doi:10.3390/ma15093048_

Round 1
Reviewer 1 Report
The abstract is too verbally it requires some quantitative results in brief
The introduction should be more professional structured as now looks rather part of report and not part of research paper; too many bullet point in ; which is author contribution – not highlighted in .
Not clear if you lap only plate surface or it works also for rounds one !
“e clamped разбить.” Please avoid Russian words !
“As can be seen from Figure 6,… by ZrO2 (12.5-22 GPa)” this should be endorsed by measurements not only linking to hardness values !
From Table 3 the lapped surface seems quite poor !
Otherwise I don’t agree taking as base of measurement of lapping the Ra other parameters should be discussed and evaluated as Rz and so on
Instead of measurement with optical microscopy other more in depth analysis is required
There are numerous results but their interpretation and discussion against literature data is almost nul
The state of art is quire out of date and novel references are required
Author Response
Dear Reviewer,
I am grateful for the helpful and interesting comments by you.
We have revised the article.
We tried to make it better.

Reviewer 2 Report
The work is very original and interesting. I have a few comments:
i. Enhance the objective in the introduction
ii. In your images, use a scale bar rather than X100 magnification
iii. Recheck your grammar editing to enhance your work.
Author Response

(The authors gave the same response as above.)

Reviewer 3 Report
The work of this manuscript seems a little interesting and potential application. In the article, the regularities of the formation of the resulting raster tool trajectories based on Lissajous figures for the lapping process of planes are established. The work is relevant and meaningful in making it possible to maximize the cutting ability of the tool, which contributes to its more uniform wear and increased productivity and processing quality. However, there are still some questions in the current manuscript, and the authors need to deeply revise the manuscript to make it more readable. The manuscript can be probably considered for publication after the revision.
1) The present and past tenses in the article are confused. I hope the author can correct them.
2) In Table 2. the Q of ASM 40/28 is 1.63, but in Figure 6. the Q changes to 1.65, hoping the author revise the value.
3) The authors need to unify the layout of figures and tables in the article
4) The abscissa name in Figure 9 is wrong, the authors need to revise it.
Author Response

(The authors gave the same response as above.)

Round 2
Reviewer 1 Report
The list of references should be improved.
Evidence of success for round surface lapping are required.
Author Response

(The authors gave the same response as above.)
